# Pathogen Dose in Animal Models of Hemorrhagic Fever Virus Infections and the Potential Impact on Studies of the Immune Response

**DOI:** 10.3390/pathogens10030275

**Published:** 2021-03-01

**Authors:** Bryce M. Warner

**Affiliations:** Zoonotic Diseases and Special Pathogens, National Microbiology Laboratory, Public Health Agency of Canada, Winnipeg, MB R3E 3R2, Canada; bryce.warner@canada.ca

**Keywords:** animal model, hemorrhagic fever, hemorrhagic fever viruses, Ebola virus, Lassa virus, pathogen dose, immune response, T cells, Th1/Th2

## Abstract

Viral hemorrhagic fever viruses come from a wide range of virus families and are a significant cause of morbidity and mortality worldwide each year. Animal models of infection with a number of these viruses have contributed to our knowledge of their pathogenesis and have been crucial for the development of therapeutics and vaccines that have been approved for human use. Most of these models use artificially high doses of virus, ensuring lethality in pre-clinical drug development studies. However, this can have a significant effect on the immune response generated. Here I discuss how the dose of antigen or pathogen is a critical determinant of immune responses and suggest that the current study of viruses in animal models should take this into account when developing and studying animal models of disease. This can have implications for determination of immune correlates of protection against disease as well as informing relevant vaccination and therapeutic strategies.

## 1. Introduction

Hemorrhagic fever viruses, defined here as viruses that can cause fever and hemorrhagic disease, include a number of viruses that are a significant cause of morbidity and mortality worldwide. Despite the name, hemorrhage does not occur in all cases and disease generally does not include excess bleeding, but is typified by fever, vascular permeability, hypovolemia, organ failure, and shock [1]. Viruses that cause hemorrhagic fevers include those of the *Filoviridae*, *Arenaviridae*, *Hantaviridae*, *Nairoviridae*, and *Flaviviridae* families. These families are diverse and contain viruses that have fatality rates that can range from less than 1% to as high as 90% in some situations. The most well-known of these viruses is likely Ebola virus (EBOV), which caused a significant outbreak in West Africa from 2013 to 2016 that resulted in more than 11,000 deaths and 29,000 cases [2]. Sporadic EBOV outbreaks have occurred throughout Africa since its discovery and even since the West African outbreak, there have been recurring outbreaks in the Democratic Republic of Congo [3]. Lassa virus (LASV) is the cause of Lassa fever (LF), one of the most common hemorrhagic fevers worldwide, with hundreds of thousands of estimated cases occurring annually in sub-Saharan Africa [4]. Since 2018, an ongoing outbreak of LF in Nigeria has resulted in hundreds of deaths [5]. The potential for outbreaks caused by these viruses remains a major concern in endemic areas due to a lack of available vaccines or therapeutic options.

There are few approved vaccines and therapeutics available for prevention and treatment of disease and outbreaks of hemorrhagic fevers still occur regularly. In late 2019, the EBOV vaccine Ervebo, a recombinant vesicular stomatitis virus-vectored vaccine expressing Ebola glycoprotein, was approved by the FDA [6]. This represented a major breakthrough in the future prevention of EBOV outbreaks. However, there are still major issues regarding vaccination of at-risk populations in Africa, from transport of vaccines to vaccine hesitancy and mistrust in authorities. A licenced vaccine against Dengue virus, Dengvaxia, targeting all four serotypes exists. However, efficacy and safety issues depending on the serostatus of vacinees have limited its widespread use in areas affected by Dengue Fever [7]. Several others have gone through some clinical testing, but there remains a need for effective preventative and therapeutic options for Dengue Fever [8]. The yellow fever vaccine is the most successful example of a vaccine targeting a virus that can cause hemorrhagic disease, and the attenuated virus vaccine has been used for decades in individuals travelling to Africa and South America [9]. Vaccine development efforts for a number of medically significant hemorrhagic fever viruses are ongoing. Treatment options for hemorrhagic fever are limited and typically consist of standard of care such as fluid and electrolyte replacement, pain management, sedation, and sometimes blood transfusions depending on the capabilities of facilities administering care [10]. Off-label use of ribavirin has been used to treat infections with several pathogens, including LASV, and Crimean–Congo hemorrhagic fever (CCHFV), though its efficacy has been debated [11,12,13]. Steroids to reduce inflammation and curtail excessive immune responses are sometimes administered and administration of immune plasma or serum has been part of treatment regimens for various viruses for years. However, no current standard treatment regimen exists for infections with hemorrhagic fever viruses. Specific immune therapies have some potential, and a cocktail of monoclonal antibodies against EBOV, ZMAPP, has undergone clinical trials for efficacy but has not been approved [14]. The very few approved options for prevention and treatment of viral hemorrhagic fevers (VHFs) highlight the need for ongoing study of their pathogenesis in animal models as well as the development of prophylactic and therapeutic options.

The study of many viruses that cause hemorrhagic fever is difficult due to most requiring a biosafety level 3 or 4 facility and many being classified as select agents in the United States. Despite these strict requirements, some viruses are widely studied and have well-established animal models of disease that have been used to examine basic aspects of infection and to test vaccines and drug candidates [15]. These models can range from small rodent models such as mice, hamsters, and guinea pigs, to more sentient options such as non-human primates (NHPs) [15]. These types of animal models are valuable tools for studying viral infections, and their pre-clinical use often precedes the advancement of medical countermeasures into human trials. An important aspect of developing vaccines is determining immune correlates of protection against infection and disease development, and animal modelling in addition to human data can be informative in this regard. Indeed, these types of studies resulted in the identification of monoclonal antibodies as a suitable therapeutic option for treatment of EBOV infection. While this value has been recognized, determining what types of immune responses are protective during hemorrhagic fever virus infections requires models of infection that are representative of the majority of human infections or those representing the most severe forms of infection. This is sometimes not achieved, as pathogen doses and routes of infection are used that may not be representative of human infections. Here I examine some of the current literature around animal models of hemorrhagic fevers, and make the case that for studying pathogen-specific immune responses and their relevance in protective immunity, a shift in how experiments are conducted may be needed.

## 2. A Primer on the T Helper Cell Subset Paradigm and Pathogen Dose

The first observation of distinct CD4 T cell populations producing different effector cytokines was made by Mossman and Coffman in the 1980s [16]. They showed that individual T cell clones could be categorized based on the cytokines that they produced, and called them T helper 1 (Th1) and T helper 2 (Th2), respectively [16,17,18,19]. At that time, Th1 cells were shown to produce IL-2, IFN-γ, GM-CSF, and IL-3 upon activation, while Th2 cells produced IL-3, BSF-1, and IL-4. Today, it is well established that Th1 cells primarily produce IFN-γ and some IL-2, while Th2 cells produce IL-4, IL-5, and IL-13 [20]. These subsets of T helper cells were shown to be involved in driving different effector functions involved in adaptive immune responses. Th1 cells aid in the development of cytotoxic T lymphocytes and M1-type macrophages, and are involved in predominantly cell-mediated immune responses. CD4 T cells that become Th2 cells are able to promote development of antigen-specific B cells into memory B cells and plasma cells resulting in predominant humoral immune responses [20]. The differentiation of CD4 T cells into either Th1 or Th2 cell types is reciprocally regulated, as the presence of Th2 cells and IL-4 inhibits the production of cytokines by Th1-type cells [19,20,21]. Since this hallmark discovery, there have been several other CD4 T helper cell subsets that have been recognized, including Th17, Th9, Th22, T follicular helper cells, and regulatory T cells [20]. Each of these cell types plays an important role in various aspects of immunity, and study into their development and regulation has been ongoing. For the purposes of this commentary, I will focus mainly on Th1 and Th2 cells and factors that drive their differentiation and how relevant animal modelling of hemorrhagic fever viruses should take these factors into account. However, it is possible that any or all of these other types of CD4 T cells may play an important role in infection outcomes, and indeed regulatory T cells have been shown to be important for the development of persistent infection in rodent hosts of different hantaviruses [22,23]. Therefore, the considerations presented here will likely have an impact on the development of pathogen-specific CD4 T cells of all subset types and detailed studies of these cells in the context of hemorrhagic fever virus infection would be of significant value.

There are a number of factors that have been shown to influence the development of CD4 T cells and that are able to drive antigen-specific Th1 or Th2 immune responses. The nature of the antigen-presenting cell (APC) that presents cognate antigen to CD4 T cells, be it a dendritic cell, B cell, or macrophage, is thought to influence Th1/Th2 differentiation [24,25,26]. There is evidence that different subsets of dendritic cells may be better at directing Th1 or Th2 immune responses [25]. The phenotype of dendritic cells, including their expression levels of various co-stimulatory molecules, can influence the type of immune response generated [27,28,29]. In this case, the site of antigenic challenge may also be an important factor tied to the type of dendritic cells and APC presenting antigen to T cells, as different types of cells are present within different compartments of the body [30]. Increasing the foreignness of an antigen can increase the number of CD4 T cells responding to linked epitopes, resulting in a bias toward Th2-type responses [31]. A significant amount of study has gone into the role of cytokines and their influence on developing immune responses, and classical regulation of Th1 and Th2 immunity involves the presence of cytokines such as IL-12, IFN-γ, and IL-4 that drive their induction [32,33,34,35]. To this end, the presence of other cell types such as CD8 T cells that are present during CD4 T cell activation, specifically CD8 T cells specific for the same antigen or pathogen, may directly interact with or produce cytokines that may alter the fate of the responding CD4 T cell. While the activation of CD4 T cells into mature Th1 or Th2 helper cells is a complex process that involves many factors in addition to those listed here, in the context of infection with microorganisms, the infectious challenge dose has been shown to play a significant role. The interplay between antigenic dose and the cooperation and number of responding antigen-specific CD4 T cells is the critical factor that I will focus on here, with a thought toward how the dose of infectious challenge can significantly impact the immune response generated in in vivo models of infectious diseases.

Classical observations of the role of antigen dose on the type of immune response were made as early as the 1950s when Salvin immunized guinea pigs with either Diptheria toxin or the protein ovalbumin (OVA), and these animals displayed a delayed-type hypersensitivity (DTH) response to antigen [36]. This DTH response preceded the antibody response, and it was noticed that depending on the dose given, the length of the DTH response seen before progress to an antibody response would change. High doses resulted in shorter DTH responses, while progressively lower doses eventually resulted in strictly DTH in immunized animals [36]. Further studies on the relationship between antibody production and DTH responses, which can be thought of in contemporary terms as cell-mediated or Th1 responses, led to the notion that these were inversely related. Studies using doses of digested flagellin given to Wistar rats further confirmed the idea that differing doses of protein antigens led to different adaptive immune responses [37,38,39]. These early experiments were done using antigens made up of whole protein, and there are some classical studies in which peptide antigens have shown an opposite effect, whereby low doses of peptide can induce Th2-mediated antibody responses while high doses induce Th1-type responses [40,41,42,43]. However, this is not the case with all peptides, and can involve other factors such as the affinity of the peptide for MHC, and as I will describe, most models using intact antigen or infectious challenge models support the idea that lower challenge doses result in Th1 responses while higher doses induce Th2 mediated humoral immunity (Figure 1). The effect of antigen dose and route on humoral immune responses can be more difficult to evaluate. Aside from examining the magnitude of the antibody response, i.e., the total amount of antigen-specific antibody or neutralizing antibody present, there are other factors to consider. The isotype of antibody induced can be critical in determining infection outcome, and the ratio of different types of antibody can be indicative of the nature of the adaptive immune response generated [44]. Examination of IgG1/IgG2 ratios following vaccination or infection can be used as a surrogate for determining Th1/Th2 immunity. However, the ability to examine specific isotypes is an option only in certain animal species for which there are available reagents.

While the described experiments above focused on specific responses to antigens given experimentally, there is a plethora of evidence that in infectious challenge models, pathogen dose is a critical factor when it comes to the type of immunity generated and for animal survival. Examples can be found for viral, bacterial, and parasitic infections and these models can inform our knowledge as to what types of immunity are critical for protection against disease and can provide the basis for vaccination strategies. Infection of mice with *Leishmania major* parasites can result in infection that is cleared or in an infection that resembles cutaneous leishmaniasis, depending on the strain of mouse infected or the dose of parasite given. At a standard dose, different strains of mice are either susceptible or resistant to infection, and this outcome is dependent upon whether a given strain mounts a Th1, cell-mediated response or a Th2 response [45,46]. Generally, BALB/c mice are susceptible to infection while CBA mice are resistant, and a clear disparity in the type of immune response is seen between these two strains when infected with the same dose of the parasite [45,46]. Interestingly, however, if BALB/c mice, the classically susceptible strain are given lower doses of the parasite, this significantly alters their immune response, skewing it toward a DTH phenotype [45]. This response renders the mice resistant to secondary challenge. This example highlights severe changes in the nature of immune response to the same pathogen, given by the same route, only at different doses.

Similar to the studies described above, studies in mice examining the Th1/Th2 nature of immune responses to bacille Calmette Guerin (BCG) have shown a similar phenomenon. Th1 responses are critical for containing infection with mycobacteria, with mixed or Th2 responses leading to disease in individuals with tuberculosis. Immunization or challenge with BCG in mice can lead to a stark difference in the Th1/Th2 phenotype of BCG-specific CD4 T cells induced by this challenge [47]. Inoculation with relatively low doses results in strictly Th1 responses with significant IFN-γ production and high IFN-γ:IL-4 ratios, while high dose inoculation leads to a mixed response [47,48]. As with *L. major* infection, the route of BCG administration is independent of the type of immune response that is generated. These observations can have implications for vaccination, whereby individuals can be administered appropriately high or low doses depending on the nature of the immune response needed for protection against exposure. Indeed, this has been shown experimentally for *L. major* infections in mice and been suggested as a strategy for prevention of tuberculosis [46,48]. Interestingly, in a mouse model of candidiasis, mice that received low doses, 10^2^ or 10^3^
*Candida albicans* cells, had higher DTH responses than those that received higher doses [49]. This DTH was not able to protect against secondary challenge with a lethal dose of the yeast unless mice were given treatments with anti-IL-4 antibody. This treatment early during initial infection impairs the ability of IL-4 to induce Th2 responses, and these mice mount protective Th1 responses against secondary lethal challenge [49]. *Leishmania major*, fungal, and various mycobacterial species are complex organisms, at least antigenically, compared with many viruses, including most that are responsible for causing hemorrhagic fevers, which are typically negative-sense RNA viruses that encode only a few proteins. The induction of immune responses in individual scenarios could vary depending on the nature of the infectious agent. However, there are examples of the dose of virus impacting the nature of the CD4 T helper response.

The immune response mounted by neonatal individuals is characterized by a significant bias toward Th2 immunity and strong IL-4-dependent induction of humoral responses [50]. While early on it was thought that newborn individuals were tolerant to most antigenic challenges, it was discovered that in fact they are immunocompetent and T cells from newborn individuals could respond immunologically in vitro and in vivo, and that this was dominated by Th2 responses. There are a number of factors that have been shown to contribute to the Th2 bias of neonatal immunity, and I will not go through those here, but only mention this brief background as a primer for the following example. Sarzotti et al. in the 1990s were able to show that while newborn mice typically generate a Th2 response to viral challenge, the dose of virus given was a critical factor in determining the Th1/Th2 nature of the immune response [51]. In mice, newly born individuals challenged with 1000 PFU of murine leukemia virus (MLV) results in rapid replication of the virus in the spleen and brain, without detectable anti-MLV CD8 T cell responses and IFN-γ production [51]. Infection of 21-day-old mice with the same dose results in strong CTL induction and protection from infection. Surprisingly, newborn mice given doses of 0.3 or 1 PFU of MLV mounted similar responses to those seen in older animals given 1000 PFU. This was evidence showing that newborn individuals are capable of mounting strong Th1, CTL responses under certain circumstances, namely viral dose. Adoptive transfer of splenocytes from newborns given the low dose into newly infected newborns given a neuropathogenic dose was able to protect against disease, suggesting that a primed immune response from mice given a low dose provides protection against lethal challenge. Another example of the dose of virus playing a key role in the development of immunity is in experimental infections of macaques with simian immunodeficiency virus (SIV). Those infected with lower doses showed T cell-mediated ability to clear SIV infection and an inability to support SIV replication in vitro in peripheral blood cells taken from those animals, without elimination of CD8 T cells prior to culture [52,53]. It is clear that the dose of antigen, be it inert protein antigens or those from infectious microorganisms, is a critical determinant of the Th1/Th2 nature of the immune response generated, and that this has significant implications for protection against disease. Furthermore, the study of these types of responses in appropriate animal models of disease can indicate what type of response can provide protection against challenge and inform vaccination and therapeutic strategies.

While there are several factors that have been shown to contribute in some fashion to the Th1/Th2 fate of CD4 T cells as outlined briefly above, many are hard to reconcile fully or in part with the data showing that changes in antigen dose result in distinct immune responses. If two different doses of antigen or pathogenic challenge are given via the same route, but elicit distinct immune responses as determined by examining CD4 T cell phenotypes, a number of those hypotheses put forward cannot explain this difference. One possible explanation is that the number of antigen-specific CD4 T cells activated upon recognition of antigen determines, in part, the Th1/Th2 nature of the immune response. This “threshold hypothesis” has been described in detail previously, and both in vitro and in vivo evidence of its effects have been characterized [54,55,56,57,58,59]. Briefly, this idea is built upon evidence showing that cooperation between antigen-specific CD4 T cells is required for their activation, following recognition of cognate antigen presented by APC, and suggests that a low number of responding CD4 T cells specific for antigen, with minimal CD4 T cell interactions leads to Th1 responses, while greater levels of interaction between CD4 cells leads to Th2 responses [54,55]. There is substantial evidence indicating that this is likely the case, and this phenomenon is intimately linked with antigen dose. Changing the dose of antigen, OVA, in the presence of a constant number of responding transgenic T cells alters the phenotype of those responding cells [59]. However, only in instances with a high density of responding CD4 T cells does antigen dose significantly influence the phenotype of CD4 T cells, as at low densities of responding T cells, Th1 responses are made regardless of concentration of antigen [59]. This effect is likely due to the number of interactions of CD4 T cells through co-stimulatory molecules such as B7/CD28 while recognizing the antigen in the context of MHC-II. These interactions involve a complex interplay of the amount of antigen present and the number of cells. While these types of experiments can unveil important mechanisms of T cell activation and function, their relevance in in vivo challenges is not fully clear, though many of these patterns remain constant in both in vitro and in vivo systems. Regardless, it is clear that however antigen dose helps determine, in part, the Th1/Th2 nature of the immune response, this is a critical factor to consider in all aspects of immunological research as well as in examining immune responses the context of infectious disease challenge models.

## 3. Important Animal Models of Viral Hemorrhagic Fever

The study of viral hemorrhagic fevers has relied upon animal models for much of our understanding of disease pathogenesis, correlates of protection, and the assessment of vaccines and therapeutics. These models can be especially important for infections that are rare in human populations for developing a deeper understanding of infection with these viruses. The use of these animal models of disease is indispensable for pre-clinical testing of vaccines and therapeutics, where the developmental track often includes the demonstration of efficacy in pre-clinical models, or the so-called FDA animal rule [60]. For many viruses, both smaller animal models as well as non-human primate models exist, often with specific species offering certain advantages and disadvantages, including their susceptibility to natural infection with wild-type viruses without the need for adaptation. This can lead to challenges in situations in which wild-type viruses cannot infect wild-type hosts of given species, for example mice or guinea pigs, and require adaptation. There has recently been a push toward reducing the use of species-adapted viruses when testing specific vaccines and therapeutics, as these may not reflect the nature of infections with non-adapted viruses. However, the alternative of using animals that are immunocompromised in the development of suitable animal models using wild-type viruses presents its own host of challenges in context of these studies, particularly when studying pathogenesis or host responses to infection. There is some trade-off when using either approach, and balanced use of both is likely needed for continued research into vaccines and therapeutics such as monoclonal antibodies or even antiviral drugs and research into mechanisms of antiviral immunity and host responses. Even confirmation of efficacy in pre-clinical animal testing in models utilizing both host-adapted viruses and wild-type viruses in genetically modified hosts is warranted. Regardless, a wide variety of animal models using different approaches have been developed for several viruses that can cause hemorrhagic fever. Here I will briefly describe some of the important models that have been characterized, with a particular focus on those that have been used for pre-clinical drug and vaccine evaluation and assessment of host immune responses.

### 3.1. Filoviruses

The *Filoviridae* family consists of three genera, the two most important of which are *Ebolavirus* and *Marburgvirus* [61]. The *Ebolavirus* genus consists of six member species, which all differ in some degree in their capacity to cause disease in humans. The most significant and well-known virus is EBOV, formerly *Zaire Ebolavirus*, the cause of most major outbreaks of disease in Africa. However, animal models have been described for *Sudan ebolavirus* (SUDV), *Bundibugyo ebolavirus*, and *Reston ebolavirus* [62]. Animal models of both Marburg virus (MARV) and Ravn virus, both members of the *Marburgvirus* genus, have been described as well [62].

A large number of studies of filoviruses utilize mouse models of infection due to the low cost, ease of handling, and availability. Since mice are not susceptible to infection with wild-type filoviruses, their adaptation to mice is required to develop suitable models. Adaptation of the virus leads to highly lethal variants of these viruses that can be used to assess the efficacy of vaccines and therapeutic drugs. Mouse-adapted EBOV infection results in several characteristics that are similar to human disease, including replication in the liver and spleen, coagulopathic dysfunction, liver and kidney dysfunction, and targeting of macrophages and mononuclear phagocytes upon infection [62,63]. A limitation of mouse models of filovirus infections is that they generally require intraperitoneal (IP) infections, a route that leads to rapid dissemination of virus, but one that is not relevant to natural infection with these viruses, which are typically contracted via mucosal surfaces or breaches in the skin coming into contact with virus in contaminated bodily fluids [64]. Pre-clinical studies in mouse models using mouse-adapted viruses have been used for a variety of vaccines and treatments for EBOV and MARV, and have led to important observations and discoveries, including in the use of the VSV-EBOV vaccine and monoclonal antibodies for treatment of Ebola virus disease [65,66,67,68]. Due to the need for adaptation of viruses to mice, immunocompromised mice, such as those deficient in the interferon alpha/beta receptor (IFNAR-/-), interferon alpha/beta/gamma (IFN-α/βγR-/-) receptors, or signal transducer and activator of transcription 1 (STAT1) have been used as models for filovirus infection [69,70]. These mice are susceptible to wild-type viruses that have not undergone adaptation, which can be valuable for performing animal experiments on newly isolated viruses or variants for which mouse-adapted viruses have not yet been generated, or for studying countermeasures against non-adapted viruses.

Guinea pigs have also been used extensively as a model for filovirus infections. As with mice, infection of guinea pigs requires adapted viruses for lethal disease to develop. Guinea pigs offer a larger animal species to study disease progression compared with mice and are readily available, but do not offer the same advantages in terms of detailed immunological studies. Its main utility is as a secondary confirmative model in addition to mouse models. Guinea pig models of EBOV, MARV, and SUDV all exist and infection results in uniform lethality [71,72,73]. Another option for a small animal model other than mice is Syrian hamsters. Mouse-adapted EBOV readily infects Syrian hamsters, resulting in disease and death within 5 days of infection [74]. Recently, a hamster-adapted MARV was reported, providing another suitable model to examine disease [75]. Ferrets represent another viable option for EBOV, SUDV, and Bundibugyo virus infections, and they recapitulate hallmarks of humans disease when infected with wild-type viruses, something that gives them an advantage over other small animal models [62,76]. Ferrets are typically more expensive, and require more specialized caging, handling, and care and so their use is often limited. All three species described here have their advantages and disadvantages, including showing signs that are consistent with human infection, but they all lack the wide availability of reagents for studying immunity outside of basic humoral responses. The development of more species-specific reagents is ongoing. However, this is a critical issue when deciding whether to study certain aspects of pathogenesis in certain models.

Non-human primate (NHP) models of filovirus infection are the gold-standard models, and closely recapitulate aspects of human infection. Infection does not require adaptation, and a wide array of reagents are available for studying the full disease course and immunological response to infection in macaque species. Several different primate species have been used for infection models including African Green monkeys, marmosets, rhesus macaques, and cynomolgus macaques [62,70]. While NHPs remain the gold standard for studying aspects of disease progression as well as testing of vaccines and therapeutics, their use is limited by cost, specialized housing requirements, and the fact that typically only a small numbers of animals can be used for a given experiment. Additionally, due to the sentience of NHPs, there are ethical limitations to their use. Most advanced studies of vaccines or therapeutics developed against filoviruses have been done in rhesus or cynomolgus macaques models, and these are generally needed before advancement into human trials. A large number of vaccine and therapy experiments against EBOV, SUDV, and MARV have been reported using NHP models of infection [77,78,79,80]. For the majority of infection models in NHPs, IM infection is performed, ensuring systemic spread of the virus and rapid disease progression. Mucosal challenges are rare, and typically require a higher dose, and some transmission experiments have been performed, with mixed results [62,81,82,83]. Overall, the use of NHPs is well established and remains the best model for studying pathogenesis and medical countermeasures. However, there remain caveats to their use when compared with the majority of human infections.

### 3.2. Arenaviruses

Arenaviruses consist of two groups of rodent-borne viruses, Old World Arenaviruses, including LASV and others found in Africa, and New World Arenaviruses found in the Americas, and include several viruses that can cause hemorrhagic fever found in South America. While LASV is the most common cause of hemorrhagic fevers worldwide, several South American Arenaviruses have been discovered that can cause hemorrhagic fever disease, including Machupo virus, Junin virus, Sabia virus, Chapare virus, and Guanarito virus [84]. While LASV is the most well-studied hemorrhagic fever-causing Arenavirus in terms of animal model development, models have been developed for some of the South American viruses that have been useful for studying disease and countermeasure efficacy.

Many models of LF have been developed and characterized, and many have been useful for testing vaccines and therapeutics. Similar to filoviruses, wild-type, immunocompetent mice are not susceptible to LASV infection. However, IFNAR-/- and STAT1 knockout mouse models have been reported [85,86,87]. When challenged with LASV, IFNAR-/- mice produce high viral titers but do not succumb to infection [85]. Chimeric IFNAR-/- mice with a wild-type immune system and STAT1 knockout mice are options for fully lethal models of LASV infection, however their use has been limited [86,87]. Guinea pigs have been the most widely used small animal model of LASV infection, with both outbred Hartley guinea pigs and inbred strain 13 guinea pigs offering viable options to study pathogenesis and test medical countermeasures [88]. Multiple strains of LASV can infect strain 13 guinea pigs, with varying levels of lethality, while adaptation is required for lethal infection of Hartley guinea pigs [89,90]. Pathogenesis of LASV infection in guinea pigs has some characteristics similar to human infection. However, it also has significant respiratory and cardiac involvement, which are typically not observed in LF in humans. Nevertheless, guinea pigs have been used extensively for assessing the efficacy of various therapeutic options and vaccines against LASV [4,88,91]. Non-human primate models are the gold standard for studying LASV infection and fully lethal infection models in both rhesus and cynomolgus macaques have been used for studying disease pathogenesis and for therapeutic and vaccine testing [88]. In addition to macaques, other primate species such as marmosets and capuchin monkeys have been used [88,92]. Infection of NHPs with LASV strongly recapitulates human disease, with several hallmarks of infection including hepatic, adrenal, and splenic necrosis as well as nervous system involvement. These models have been well characterized, and any significantly advanced treatment or vaccine developed against LASV will likely undergo testing in one NHP model or another before advancement into human trials.

Of the South American arenaviruses that are important for human disease, several animal models have been reported. Lethal infection of guinea pigs, either strain 13 or Hartley, with Junin, Machupo, Lujo, or Guanarito virus provides an adequate model for studying these pathogenic New World arenaviruses [93,94,95,96]. These infection models show severe disease manifestations including viral deposition in the lymph nodes, hemorrhaging in the gastrointestinal tract, necrosis in multiple tissues, as well as some nervous system involvement. Once again, NHP models have been described for these viruses as well, including macaque, marmoset, and African green monkey models. For Junin virus, severe disease occurs in rhesus macaques as well as in marmosets, and these models have been used to assess the efficacy of treatments such as ribavirin and cross-protective immunity among New World arenaviruses [97,98,99,100]. Infection of rhesus macaques with Guanarito virus is not lethal, though animals do support viral replication and do show signs of illness [101]. Machupo virus infection can cause a uniformly lethal infection in rhesus macaques, but infection of African green monkeys is not fully lethal [102,103]. Regardless of lethality, these represent viable models for studying these viruses that can often be overlooked relative to some of the others mentioned above.

### 3.3. Crimean–Congo Hemorrhagic Fever Virus

Crimean–Congo hemorrhagic fever virus (CCHFV) is a member of the *Nairoviridae* family or the Bunyavirales order and is the causative agent of Crimean–Congo hemorrhagic fever (CCHF). It is a tick-borne pathogen that has a wide geographic distribution, from Eastern Europe, through the Middle East and Asia, and into Africa [104]. The virus readily infects a number of species that are commonly used as livestock, and infections of human can occur through exposure to infected ticks or infected animals. Detailed pathogenesis studies of CCHF have previously been lacking due to few animal models of viral infection. Many species can be infected with CCHFV, but they do not show signs of infection or illness despite some showing high viremia and levels of antibody [105,106]. Immunocompetent mice are not susceptible to infection as adults, so neonatal mice have been used as an infection model. However, this model does not recapitulate aspects of human disease and has practical limitations [107,108]. More recently, STAT-1 and IFNAR-/- mice have been used as a model of infection. These models result in lethal infection and show high levels of viremia and viral dissemination in multiple tissues [109,110,111]. These models also show characteristics such as elevated liver enzymes, lymphocyte depletion, and lesions in the liver which can be present in human infection. Recently, an immunocompetent mouse model of CCHF infection was reported, in which a mouse-adapted virus was able to cause disease in several strains of mice, as well as lethal infection [112]. While wild-type hamsters are not susceptible to infection with CCHFV, a recent study showed that STAT-2 knockout hamsters succumb to infection and can be used as a model of lethal CCHF disease. This model, similar to the immunocompromised mouse models described above, involves histopathological lesions in the liver, along with altered hematological and biochemical parameters indicative of liver and kidney dysfunction [113]. It can also be used to evaluate the efficacy of antiviral therapies such as ribavirin [113]. This provides another small animal model suitable for studying pathogenesis, though alterations in the innate immune responses in these immunocompromised animals may significantly impact studies of the immune response to infection, and this should be a consideration when using these animals for studying viral infections. The best model for studying CCHF disease may be the cynomolgus macaque model, which can result in up to a 75% fatality rate depending on the route of infection [114]. This model showed significant hemorrhaging in infected animals along with high levels of viremia and liver and kidney dysfunction. This model also provides an opportunity to examine the immune response in infected individuals, which has been characterized by examining the serum levels of various cytokines and chemokines. Since being first reported, the cynomolgus macaque model has been used to test the efficacy of antiviral therapy and vaccination, and the spectrum of disease can vary significantly [115,116]. While the initial characterization of the model reported up to 75% lethality, further studies have shown mild to moderate disease, including the persistence of viral antigen and RNA in the testes as late as 28 days post-infection [117]. Infection with multiple CCHF strains was also shown to cause mild disease, suggesting that the lethality in the macaque model is variable, but likely represents a wide clinical spectrum that is evident in human infection [118]. Overall, the macaque model has proven beneficial for efficacy testing of therapeutics and vaccines despite reduced lethality and will likely remain an important tool for CCHF research.

### 3.4. Rift Valley Fever Virus

Rift Valley Fever (RVF) is a viral disease of ruminants in Africa that can have a severe impact on livestock, and is characterized by fever, hyperpnoea, and inappetence in newborn livestock and by abortion in female animals, and is caused by Rift Valley Fever virus (RVFV) [119]. While infection of livestock is of critical importance, human infection can also occur and has a variable outcome, ranging from little or mild disease to possible hemorrhagic disease in severe cases [119]. Typical flu-like illness can occur before more severe manifestations such as hemorrhage and encephalitis. Many animal models of RVFV infection exist, including mouse, rat, hamster, and NHP models, with varying degrees of lethality and resemblance of features of human disease [119]. Mice are susceptible to RVFV infection, which results in mortality and clinically resembles human infection in important ways [119,120]. In mice, hepatitis and encephalitis are the primary outcomes of RVFV infection, and the liver tends to be the main target early following infection. Extensive hepatocyte damage occurs along with elevated liver enzymes and bilirubin and this acute damage leads to death in most cases. In animals that have prolonged disease, encephalitis occurs, and this may also even occur in mice that have been treated and survive acute infection [119]. The ability to study infection in wild-type strains such as BALB/c, and the ability to use genuine, non-adapted virus, allows for the ability to test pathogenesis of novel isolates and strains, a highly valuable trait of a given model. Rats are also susceptible to RVFV depending on the strain, and similar to mice, can succumb to infection very early after exposure, within 3–5 days in some cases, even following low dose infection [119,121,122]. Additionally, similar to mouse infections is the viral hepatitis that occurs along with high viremia. Fulminant hepatitis along with severe pathological lesions and necrosis in the liver lead to death. Strains such as ACI and MAXX rats have reduced mortality following infection, and inbred Lewis rats do not show clinical disease, offering a wide range of infection outcomes in the rat model to study similar infection outcomes in humans. Very early reports of RVFV infection of NHPs demonstrated that infection was not universally lethal as in some rodents. However, animals did show clinical signs of infection, and outcome depended on the strain of virus and species of primate used [119,123,124,125,126]. Rhesus macaques infected with a high dose of virus intravenously likely represent the best model of infection in NHPs, with a range of outcomes that mimic what is seen in human infections. Animals may succumb to infection following a severe disease course characterized by anorexia, depression, vomiting, weakness, and hemorrhaging. Severe disease also involves liver necrosis and disseminated intravascular coagulation. Some may experience severe illness and survive, while others do not develop the characteristic signs of infection and have mild to no illness. While there may be some value in having this range of outcomes, the large portion of animals that survive and do not experience illness makes assessing the efficacy of vaccines and therapeutics difficult. The use of intravenous exposure also may not be a biologically relevant infection route, and may have effects on infection outcome in vaccinated animals.

### 3.5. Flaviviruses

The family *Flaviviridae* is a large family of viruses that consists of more than 70 viruses. Those that can cause hemorrhagic disease are a part of the *Flavivirus* genus and are mainly carried by arthropod vectors. These include Dengue virus, yellow fever virus, Kyasanur forest disease virus, and Omsk hemorrhagic fever virus. Here, some of the animal models for these viruses will be described briefly.

Dengue Fever, caused by Dengue virus (DENV) infection transmitted by mosquitos, is a disease that is typically mild and self-limiting, but can sometimes result in hemorrhagic disease, called Dengue hemorrhagic fever [127]. This hemorrhagic disease occurs in approximately 250,000 people each year and can result in severe thrombocytopenia, capillary leakage, and shock [127]. There are few animal models of DENV infection, with most not resembling human disease. Various NHP species do not show any clinical signs of illness though they can become viremic and do seroconvert [128,129,130,131]. Rhesus macaques are susceptible to infection with all four serotypes of DENV. However, they do not show overt signs of disease, only altered biochemical, hematological, and coagulation parameters [132]. The lack of a suitable and reliable NHP model of DENV means that many studies utilize various mouse models of DENV infection. However, these models present their own limitations. While wild-type mice can be infected, they do not show disease, with limited viremia, therefore a number of mouse models of DENV have been developed. These include various immunocompromised strains, including IFNAR-/- and AG129 mice, as well as mice which have IFNAR deletion in specific cell types, which can be infected with mouse-adapted strains to cause severe disease [133,134,135]. A number of other immunocompromised or chimeric strains of mice such as NOD-SCID, SCID-PBL, NOD/SCID/IL-2R-/-, and STAT1-/- mice are susceptible to infection depending on virus serotype, and these do show signs of infection seen in humans including rash, fever, and thrombocytopenia [134]. Some of these mouse models have been used to assess vaccine and therapeutic efficacy, including whether antibody dependent enhancement of disease can occur following infections of DENV and other flaviviruses such as Zika virus [136]. Overall, while the development of DENV vaccines has progressed significantly in the absence of strong animal models, this is an important area in need of development for the advancement of our understanding of DENV infection and Dengue Fever.

While the yellow fever vaccine has existed for many decades, yellow fever virus (YFV) still causes hundreds of thousands of cases of illness every year and up to 30,000 deaths in populations that are unvaccinated [137]. Like other hemorrhagic fevers, illness can vary widely, with severe disease along with high viral loads, liver and kidney failure, hemorrhaging, and cardiac involvement [138]. There are several animal models that closely resemble yellow fever in terms of signs of illness, including hamster, mouse, and NHP models. Suckling mice have been used. However, the encephalitic disease seen is not what is typically seen in humans, and this model does not have much value [70]. A hamster-adapted virus has been developed which causes a wide range of signs following infection [139,140]. Viremia, thrombocytopenia, coagulation dysfunction, liver damage, and lymphopenia all occur in this lethal model of disease. Mouse models of yellow fever have also been characterized, with IFNAR- and STAT1-deficient mice representing highly lethal models that could be used to assess efficacy of some medical countermeasures [141]. While some species of NHP do not develop disease following YFV infection, rhesus and cynomolgus macaques are susceptible to infection, and present a disease course similar to that seen in humans, with only a more rapid time to death [142,143,144]. Macaques develop hepatitis, jaundice, renal failure, coagulopathy, and shock with high viral loads. NHP models may represent the best opportunity to further study yellow fever pathogenesis, or to test newly developed treatment regimens.

Two other hemorrhagic fever viruses that are not often discussed are Kyasanur forest disease virus (KFDV), and Omsk hemorrhagic fever virus (OHFV), which can cause Kyasanur forest disease and Omsk hemorrhagic fever, respectively. These viruses are both tick-borne pathogens whose primary reservoirs are rodents, and both have been relatively under-studied and lack reliable animal models of disease. KFDV infection of some different species of monkeys results in disease. However, these species are not commonly available [145,146]. Infection of juvenile BALB/c mice results in lethal infection, and a comparison of KFDV infection of C57Bl/6J mice with that of another related virus, Alkhumra virus, showed that KFDV infection is lethal [70,147]. Other small animal models of KFDV infection have been attempted. However, infection of guinea pigs, hamsters, and ferrets did not result in clinical signs, though there was detectable virus replication in guinea pigs and hamsters [148]. OHFV can cause some signs of disease in its rodent hosts, typically muskrats. However, very few experimental models have been reported. Mouse models using both BALB/c and C57Bl/6 mice have been described, with lethal infection possible, and some signs of infection mimicking those seen in humans [149,150]. However, animal model development for both of these pathogens has been lacking, and descriptions of pathologic mechanisms or immune system involvement or protection against infection have not been made. These represent just two viruses out of many understudied viruses for which animal model development could progress in the future, where researchers could utilize appropriate routes and doses of infection to examine specific aspects of infection that could inform therapeutic and vaccine develop or clinical treatment of disease.

### 3.6. Hantaviruses

Hantaviruses, like CCHFV, are part of the Bunyavirales order, and are found worldwide. Depending on location, they can cause two diseases in humans, called hantavirus cardiopulmonary syndrome and hemorrhagic fever with renal syndrome (HFRS). HFRS is caused by so-called Old World hantaviruses found in Europe and Asia, and tens or even hundreds of thousands of cases can be reported each year, with a case fatality rate that ranges from less than one percent to as high as 15% depending on the causative pathogen [151]. HFRS pathogenesis typically involves renal dysfunction, but can include some hemorrhagic manifestations as well as vascular leakage, and was originally named Korean hemorrhagic fever following its description in the 1950s [151,152]. There has been a focus on infection models in natural reservoir host species for hantaviruses, and animal models of disease caused by hantavirus infection are relatively limited, particularly HFRS models. There are limited models of HFRS disease, though several infection models in different species with viruses that cause HFRS have been reported. Early studies showed that infection of suckling mice with Hantaan virus is lethal, though this is not a practical or relevant model for studying HFRS, and infection of progressively older mice significantly reduces mortality [153]. Syrian hamsters have been used as a model for hantavirus infections, and this is one of the standard lethal models of Andes virus infection [154,155]. Infection of hamsters with Puumala, Dobrava, Seoul, or Hantaan viruses results in viral replication and animals do not show signs of disease [154,156]. These models have been used to test the immunogenicity and efficacy of vaccines and immune serum against infection [156,157,158,159]. Despite their utility in these settings, these models do not faithfully recapitulate HFRS pathogenesis, and models more closely resembling disease are being pursued. A recent report of Puumala and Hantaan virus infection of ferrets showed that they do lose weight following infection, but do not show any clinical signs or kidney dysfunction [156]. This same report also described an infection of marmosets with Hantaan virus, which once again showed susceptibility to infection, but no clinical signs. This model was used to show the protective efficacy of polyclonal antibodies produced in transchromosomal bovines [156]. The marmoset model is not the first use of NHPs to attempt to model HFRS, as infection of cynomolgus macaques with Puumala virus has been shown to result in mild HFRS, with viral antigen found in the kidneys of infected animals along with inflammation and tubular damage [160,161]. This model also results in hematuria, proteinuria, and a strong inflammatory response [161]. This macaque model may represent the best model of HFRS disease to date, though the cost, availability, and logistical issues of using NHPs have limited its use to only a few published studies. There remains a critical need for suitable animal models of HFRS that show signs of disease and reliably resemble human infection.

## 4. Challenge Dose Used in Animal Experiments

Of those animal models described above, one of the critical aspects is the pathogen dose, which, as covered earlier, can significantly impact the outcome of infection and the immune response generated by challenged animals. In most models, what is typically used is a very high challenge dose, on the order of 100–1000 times the median 50% lethal dose (LD_50_) (doses used in a number of published studies are listed in Table 1). Additionally, during initial model characterization, multiple routes of infection are sometimes used. However, ultimately, a specific route is often decided upon for ease and uniformity, in many cases IP for rodents and intramuscular (IM) for NHPs. This can also have a differential impact on the host response and infection kinetics, with different doses having different lethality rates, mean times to death, and induction of host responses. A high dose is used in many vaccine and therapeutic studies in order to ensure death in the control group, with incrementally less lethal doses posing the risk of having survivors, which will diminish the statistical power of the performed experiments. The use of doses closer to the LD50 would require a greater number of animals in each experiment which may not gain ethical approval or be economically feasible. While I do suggest that a close look at the dose of virus used in experiments is important, a critical aspect to consider is the origin of given pathogens used in various studies. While a specific TCID_50_, PFU, or LD_50_ value may be reported, there may be considerable variation in the derivation of these numbers between different laboratories without some standard method of measuring viral titers. Differences in virus stock production, including cell types used for propagation, titer calculation methods, or methods in determining cytopathic effect could all influence the true titer. Slight differences in viral passage history or whether a virus is a product of reverse genetics could also result in minor differences in in vivo infection outcomes. Preparation methods giving rise to different amounts of defective interfering particles could alter the infectious titer as well as the host response to infection. Back-titer calculations of the true dose of virus given in animal studies are not always reported, and a drop in titer due to prolonged storage could also be important. Therefore, a direct comparison of infection models even using the same pathogen in the same host species across different groups may be tricky. It is also true that for many viruses, the infectious and lethal human doses are not known, or have not been well characterized. This makes direct comparisons with animal models difficult, and can make it hard to determine what constitutes a “high” dose compared to what a human would typically be exposed to. Moreover, how does one consider a systemic intraperitoneal injection of a dose of virus that is considered hundreds of times what is needed for lethality compared with a mucosal exposure to an unknown amount of virus contained within bodily fluids? These are some variables that make it difficult to establish direct comparisons between animal models and human infections. While these and other limitations exist, an awareness of how these factors might influence infection outcome and a wider range of studies that attempt to address some of them could be valuable for many pathogens.

There are several reports of a low dose or a wide range of doses in animal models, and these have led to important insights into disease pathology or host responses. Surprisingly, a higher dose of EBOV in mice can lead to higher levels of survival in certain strains [162]. These types of experiments or studies used to determine the LD_50_ for given pathogens in specific animal species may often be the only published studies that report the outcome after challenge with a wide range of doses or routes that includes low doses, and these will often only examine lethality and not include a detailed examination of pathogenesis or host responses. However, as discussed above, there are reasons why this type of analysis might be important. Lower, non-lethal doses may lead to protective immunity that can be characterized, providing valuable insight into how to prevent disease against a given virus. Experiments in which a dose around the LD_50_ is given can result in both surviving and non-surviving animals, providing a chance to examine key aspects of pathogenesis and disease development, as well as immune responses. Even in some models, a high dose, many times greater than the LD_50_, can result in disparate outcomes, as seen in EBOV infection of collaborative cross mice [162,163]. In these mice, depending on genetic background, a standard dose across dozens of genetic lines can result in a wide range in infection outcomes, from severe hemorrhagic disease to resistance. These types of studies offer the chance to study specific alleles or transcriptional signatures that are associated with disease susceptibility or resistance, such as TIE and TEK in animals experiencing lethal Ebola virus disease [164]. A study of CCHF infection in IFNAR^−/−^ mice showed that infection with different viral strains at varying doses could have different outcomes, with mice infected with the Hoti strain, surviving infection and modelling human convalescence [165]. This study also examined the immune response in infected animals, providing an indication as to the type of responses that may be correlated with protection and survival.

Different doses given via different routes may significantly change infection outcome, and this can be critical for viruses that may preferentially infect humans via mucosal routes as opposed to systemic infections resulting from IP or IM injections (Figure 2). Additionally, as with *L. major* infection, the outcome of infection in different strains of mice can differ significantly. The underlying biology of specific strain differences can help determine what types of immunity may be important for protection against disease development or lethality. Indeed, in a recent report on the development of a lethal SARS-CoV-2 mouse model, infection of BALB/c mice and C57BL/6 mice resulted in significantly different lethality and disease outcomes [166]. In addition, viral infection in differently aged mice with the same dose can result in significantly different outcomes of infection [166]. The use of high pathogen doses in studies of different therapeutic drugs that may have immunomodulatory effects could affect the mechanisms of those drugs if the dose of pathogen is disproportionately high, and the effects of that dose dilute out any beneficial effects offered by the drug treatment. Moreover, doses of virus that are too high may also significantly affect the kinetics of disease, with early time to death, and in some of the models mentioned above, the mean time to death can be less than one week. Clearly in these instances, the development of the antiviral adaptive immune response will be affected, and may not even be generated at all. In instances in which certain therapies may be dependent upon a coordinated response in tandem with the host immune system, this would not provide an adequate opportunity to examine a drug’s potential benefits. While the use of high doses in the types of experiments described is generally warranted so as to ensure lethality across groups and to assess the robustness of given protective benefits of vaccines and therapeutics, these are just some examples of things that may be important to keep in mind when studying a given virus in a given disease model. Certainly, studies that are attempting to examine host immune responses to viral challenge will benefit from experiments in which lower doses are used. This may include the need for greater numbers of animals to perform experiments in which the dose given results in a lethality rate that matches what is seen upon typical human infection. Ideally, these types of studies would be performed early on, during model characterization and even during LD_50_ determination, as this information could be invaluable in the future development of vaccines and drugs. It would be impractical to suggest that researchers studying viral hemorrhagic fevers in animal models adopt a universal change to commonly used methods for producing lethal infections. The use of current models has led to many important discoveries as well as insights into pathogen-specific immune responses, even in models that have used high doses and multiple infection routes. However, I hope to bring about a discussion of a potential aspect of animal model development that researchers can be aware of when designing experiments, particularly when studying disease pathogenesis or during initial model development.

## 5. Final Thoughts

While the mechanisms that decide the Th1/Th2 phenotype of CD4 helper T cells and the generation of different types of immunity are thought to vary, classical and contemporary evidence indicates that the dose of antigen or pathogen can have a significant influence in this regard. Critically, when utilizing animal models of viral infection, researchers often use the highest dose that is practical in order to ensure lethal outcomes. However, the dose that is used may have unintended consequences in the context of the experimental goal. As shown, there is a clear trend toward experiments that utilize these high doses, with a number of studies that include some form of analysis of the host immune response to infection. In light of evidence showing that dose can considerably alter the generation of specific types of immunity, depending on the type of study, an important examination of the effect of pathogen dose might be warranted. The host responses examined during or following infection may offer evidence as to what types of immune response may provide protection from disease. However, it may be such that doses that are too high could inadvertently skew immune responses. I hope to provide here some justification for expanded or additional studies of the viruses mentioned in suitable animal models that can help reveal more regarding the effects of the host immune response, sometimes a neglected area of viral pathogenesis studies in these models. These studies would utilize appropriate models in which varying doses during viral infection can help to reveal key aspects of naturally developed protective immunity as has previously been the case for various parasitic, bacterial and viral infections [45,46,47,48,49,50,51,52,53]. These types of studies would not act to replace valuable vaccine and therapeutic testing in which high doses are needed to confirm robust efficacy, but to supplement our knowledge of how the immune response develops during infection with these viruses. This could help form critical foundational knowledge which will lead to viable clinical countermeasures for what are often diseases that have severe clinical outcomes.

## Figures and Tables

**Figure 1 pathogens-10-00275-f001:**
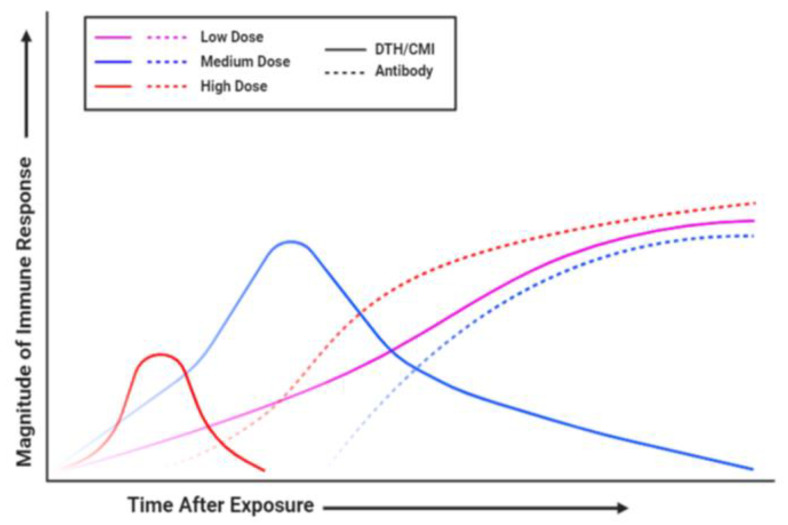
Kinetics of DTH and antibody responses to antigenic challenge. The effect of antigen dose on the kinetics of the immune response is shown. Low doses generate primarily delayed-type hypersensitivity or cell-mediated responses, while higher doses result in quicker and more robust antibody generation. Solid line represents DTH/CMI, while dashed line represents antibody responses. Magenta = low dose of antigen; blue = medium dose; red = high dose. DTH: delayed-type hypersensitivity; CMI: cell-mediated immunity.

**Figure 2 pathogens-10-00275-f002:**
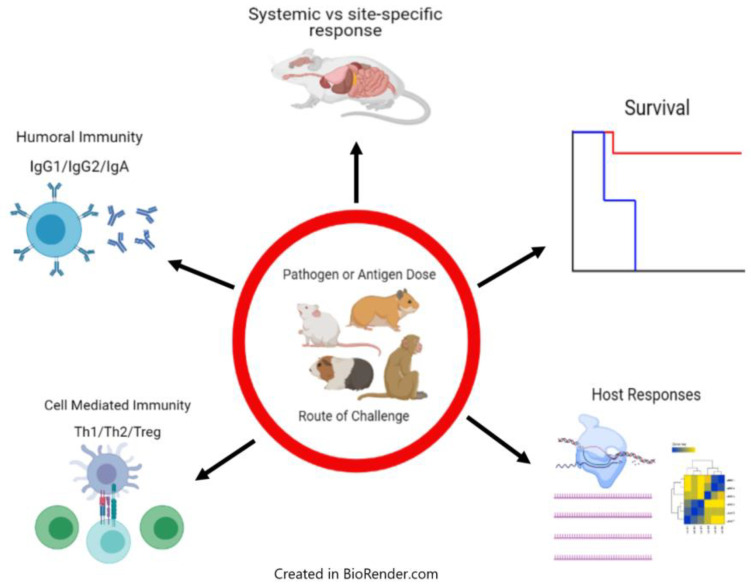
Schematic of infection outcomes and important immunological considerations following infection with hemorrhagic fever viruses. During animal models of hemorrhagic fever virus infections, including those in mice, hamsters, guinea pigs, and non-human primates, pathogen or antigen dose and route of challenge can have a significant effect on various infection outcomes, represented here. Ratios of antibody isotypes as well as the magnitude of the antibody response is dependent upon the challenge dose, with greater IgG1 production during predominant Th2 responses. IgA production may be more robust following mucosal challenge. The generation of Th1 vs. Th2 responses has been shown to be heavily dependent upon the challenge dose, with this outcome sometimes correlating directly with survival. Paradoxically, sometimes pathogen dose may be inversely related to survival, as seen in EBOV infection in certain strains of mice. Challenge route may also impact the development of site-specific pathologies and/or immune responses, seen commonly in infections with respiratory pathogens. Finally, host transcriptional responses may vary based on pathogen dose, route, viral strain, or host genetic background, revealing key pathways involved in susceptibility or resistance to infection.

**Table 1 pathogens-10-00275-t001:** Examples of Pathogen Dose in Animal Models of Hemorrhagic Fever Virus Infections.

Pathogen	Animal Species	Pathogen Dose and Route	Types of Studies	References
EBOV	Mice	10^0^/10^2^ PFU, 1000× LD50, 1000FFU; IP	Pathogenesis, Model Dev, Vaccination, Therapeutics	[63,65,66,67,167]
	Guinea Pig	1000× LD50, 10,000× LD50; IN, IP	Transmission, Pathogenesis	[168]
	Ferret	200 TCID50, 1000 PFU; IN, IM	Pathogenesis, Model Dev	[76,169]
	Hamster	1000 FFU; IP	Pathogenesis, Model Dev	[74]
	NHP	1000 PFU, 1000 TCID50; IM	Pathogenesis, Vaccination, Therapeutics, Transmission	[79,80,83,170,171]
SUDV	Mice	500–5000 PFU; IP	Pathogenesis, Model Dev	[172]
	Guinea Pig	3.55 × 10^−2^–3.55 × 10^3^ TCID50; IP	Pathogenesis, Model Dev	[73]
	Ferret	1000 PFU, 1000 TCID50; IN, IM	Pathogenesis, Model Dev	[76,173]
	NHP	1000 PFU, 50 PFU, 500 PFU; IM, Aerosol	Pathogenesis, Vaccination, Therapeutics	[80,174,175]
MARV	Mice	1000–100,000 PFU, 100 TCID50; IN, IM, SC	Pathogenesis, Model Dev	[176,177,178]
	Guinea Pig	2.32 × 10^−2^–2.32 × 10^2^ TCID50; IP	Pathogenesis, Model Dev	[167]
	Hamster	0.001 PFU–1000 PFU; IP	Pathogenesis, Model Dev	[75]
	NHP	1000 PFU, 1000 TCID50; IM	Pathogenesis, Vaccination, Therapeutics	[80,179,180,181]
LASV	Mice	10^4^ PFU, 1000 FFU; IP	Pathogenesis, Model Dev	[85,86,87]
	Guinea Pig	2–2.4 × 10^5^ PFU, 10^4^ TCID50, 10^5^ TCID50; IP, SC	Pathogenesis, Vaccination, Therapeutics	[89,90,91]
	NHP	10^5.1^ PFU, 10^6.1^ PFU, 10^4^ TCID50, 3000 PFU; SC, IM, IP	Pathogenesis, Vaccination, Therapeutics	[80,88,92]
Junin virus	Guinea Pig	4–4000 PFU, 5000 PFU; IP	Pathogenesis, Model Dev, Therapeutics	[93,182]
	NHP	10^5^ TCID50, 1000 mouse LD50; IM	Pathogenesis, Model Dev, Therapeutics	[98,99,100,183]
Machupo virus	Guinea Pig	10–1000 PFU, 2000 PFU; aerosol	Pathogenesis, Model Dev	[94,95]
	NHP	1000 PFU; SC	Pathogenesis	[102,103]
Guanarito virus	Guinea Pig	10–1000 PFU; SC	Pathogenesis, Model Dev	[96]
	NHPs	10^3.4^ PFU; SC	Pathogenesis	[101]
RVFV	Mice	0.1–1000 PFU; SC, IP	Pathogenesis, Model Dev	[68,119,120]
	Rat	10^3^ PFU, 10^5.7^ PFU; SC	Pathogenesis, Model Dev, Vaccination	[121,122]
	Hamster	1–100,000 PFU, 150 PFU, 30 PFU; SC, aerosol	Pathogenesis, Model Dev, Vaccination	[184,185,186]
	NHP	10^6^ mouse LD50,10^4.7^ PFU, 10^5.3^ PFU; SC	Pathogenesis, Model Dev, therapeutics	[120,187]
CCHFV	Mice	10^2^ PFU, 10^3.5^, 10^1^-10^6^ PFU, 10^4^ TCID50; IC, IP	Pathogenesis, Model Dev	[105,106,107,108,109,110]
	Hamster	1–10,000 TCID50; IP, IM, SC	Pathogenesis, Model dev, Therapeutics	[114]
	NHP	10^5^ TCID50; SC, IV, SC/IV	Pathogenesis, Model Dev	[115,116,117,118]
DENV	Mice	5 × 10^6^ PFU, 2 × 10^5^ FFU, 10^6^ FFU; IP, IV	Pathogenesis, Model Dev	[135,136]
	NHP	2 × 10^4^ PFU, 10^5^ PFU; SC, IP	Pathogenesis, Model Dev, Therapeutics	[128,130,131]
YFV	Mice	10^4^ PFU; SC	Pathogenesis	[141]
	Hamster	10^6^ TCID50; IP	Pathogenesis, Model Dev	[139,140]
	NHP	1000 mouse LD50, 1000 PFU; SC	Pathogenesis, Therapeutics	[142,143]
Hantaviruses	Hamster	0.2–20,000 PFU (HTNV, PUUV, SEOV); IN, IM, IP	Pathogenesis, Therapeutics, Vaccination	[156,157,158,159]
	Ferret	94,000 PFU, 200,000 PFU, 164,000 PFU; IM	Model Dev	[156]
	NHP	10^5^ vole ID50; IV	Pathogenesis, Model Dev	[160,161]

Abbreviations: IP, intraperitoneal; IM, intramuscular; IN, intranasal; SC, subcutaneous; IC, intracranial; IV, intravenous; Dev, development; HTNV, Hantaan virus; PUUV, Puumala virus; ANDV, Andes virus; SEOV, Seoul virus.

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
