# Peer review of "Pathogen Dose in Animal Models of Hemorrhagic Fever Virus Infections and the Potential Impact on Studies of the Immune Response"

_pathogens, 2021, doi:10.3390/pathogens10030275_

Round 1

Reviewer 1 Report

This paper by Warner presents a review about an important and timely issue, namely how the viral dose use in the different animal models of infection with hemorrhagic fever (HF)-causing viruses can have an important impact in the host immune response to infection and thereby in the outcome of the infection. Likewise, the review brings to the reader’s attention how the kinetics and amount of antigen expression delivered via vaccination can have also important implications for the quality and magnitude of the triggered immune response.

The review is clearly written and structured in three sections. In the first one the author discusses how the pathogen dose can influence the type of CD4 T helper responses, which has critical implications for the type of adaptive immune response raised by the infected host. In the second section, the author reviews the most relevant animal models of viral HF. The third and last section is focused on the key issue of the review, the influence of the pathogen dose on the course and manifestations of infection in the corresponding animal model.

The first section of the review would benefit from the incorporation of table providing information about the key immunological parameters that should be measured to assess the magnitude of the effect of pathogen, or antigen, dose, and route of its delivery, on changes in the host immune response profile.

The final section of the paper addresses the central question of this review on how the pathogen dose can influence the quality and magnitude of the host immune response and outcome of infection. The author has made some valid and useful points that should be taken into consideration for the design of experiments examining the pathogenesis of HF viruses and vaccine and antiviral drug efficacy against these agents. The issues of how the host genetics (different strains) and age influence the immune response to pathogens have been extensively documented in the literature and the present review does not add any novel insights on these issues. More relevant is the discussion about considering the pathogen dose in challenge experiments studying either the pathogenesis of the virus or protection conferred by vaccination or therapeutics. I concur with the author that most, if not all, currently use animal models of HF viral diseases tend to use a rather high dose of the pathogen, which may not accurately recreate what happens during natural exposure to the pathogen. However, as the author indicates, this approach is usually dictated by the large costs, unaffordable in most cases, associated with the need of using large numbers of animals if the challenge dose is of a magnitude resulting only in a % of animals developing disease symptoms, or fatal outcome. The author has identified a real and important issue, but I do not think he did provide a clear workable solution to this problem.

One aspect that would be important to discuss further is how pathogen dose is assessed. Comparing pathogen dose between experiments done by different laboratories is not an easy task as determination of the corresponding viral titers may have been done using different assays without the incorporation of a standard that could be used by the different laboratories. Likewise, the procedures used for preparing of viral stocks can differ between laboratories resulting in stocks with different content of defective interfering particles that could significantly influence the host immune response and course of infection.

Author Response

This paper by Warner presents a review about an important and timely issue, namely how the viral dose use in the different animal models of infection with hemorrhagic fever (HF)-causing viruses can have an important impact in the host immune response to infection and thereby in the outcome of the infection. Likewise, the review brings to the reader’s attention how the kinetics and amount of antigen expression delivered via vaccination can have also important implications for the quality and magnitude of the triggered immune response.

The review is clearly written and structured in three sections. In the first one the author discusses how the pathogen dose can influence the type of CD4 T helper responses, which has critical implications for the type of adaptive immune response raised by the infected host. In the second section, the author reviews the most relevant animal models of viral HF. The third and last section is focused on the key issue of the review, the influence of the pathogen dose on the course and manifestations of infection in the corresponding animal model.

The first section of the review would benefit from the incorporation of table providing information about the key immunological parameters that should be measured to assess the magnitude of the effect of pathogen, or antigen, dose, and route of its delivery, on changes in the host immune response profile.

Thank you for the suggestion. A figure has been added to this effect. And a brief discussion put into the text discussing these issues.

The final section of the paper addresses the central question of this review on how the pathogen dose can influence the quality and magnitude of the host immune response and outcome of infection. The author has made some valid and useful points that should be taken into consideration for the design of experiments examining the pathogenesis of HF viruses and vaccine and antiviral drug efficacy against these agents. The issues of how the host genetics (different strains) and age influence the immune response to pathogens have been extensively documented in the literature and the present review does not add any novel insights on these issues. More relevant is the discussion about considering the pathogen dose in challenge experiments studying either the pathogenesis of the virus or protection conferred by vaccination or therapeutics. I concur with the author that most, if not all, currently use animal models of HF viral diseases tend to use a rather high dose of the pathogen, which may not accurately recreate what happens during natural exposure to the pathogen. However, as the author indicates, this approach is usually dictated by the large costs, unaffordable in most cases, associated with the need of using large numbers of animals if the challenge dose is of a magnitude resulting only in a % of animals developing disease symptoms, or fatal outcome. The author has identified a real and important issue, but I do not think he did provide a clear workable solution to this problem.

I want to thank the reviewer for their comments. I have attempted to address this issue somewhat, though I do agree that there is a lack of a clear solution to this problem. In the text it is suggested that this issue should at least be considered during initial animal model development and acknowledged, even if it is likely impractical to expect researchers to alter their typical animal models, specifically for testing drugs and vaccines. My main goal was to raise this issue and hope that is at least brings about a discussion of how this might impact animal studies and their outcomes.

One aspect that would be important to discuss further is how pathogen dose is assessed. Comparing pathogen dose between experiments done by different laboratories is not an easy task as determination of the corresponding viral titers may have been done using different assays without the incorporation of a standard that could be used by the different laboratories. Likewise, the procedures used for preparing of viral stocks can differ between laboratories resulting in stocks with different content of defective interfering particles that could significantly influence the host immune response and course of infection.

I want to thank the reviewer for pointing this out, and it was also brought up by reviewer 2, and is a valid point. I have added some points to the text to reflect that this is often the case, and is an important consideration, and that this needs to be considered when evaluating the data presented in animal model papers.

Reviewer 2 Report

In the manuscript by Warner et al. the author reviews the effect of antigen dose on development of immune responses, summarizes current animal models for VHF and ends with a discussion on challenge dose and model utility. Overall the paper is well written and has extensive references. However, the later parts of the manuscript are entirely speculative and not well referenced. Although I happen to agree with the author's conclusions that challenge dose is an important consideration for model development and as a further extension that severe but non-lethal models are a useful tool to study immune responses the later portions of the manuscript need revision.

Major comments:

Author focuses on T cells in section 2, what about humoral responses?

With regards to section 3 on CCHFV, the development of an immunocompetent mouse model for CCHFV has been recently reported and should be included in this section.

Line 453 - 462: The limitations of this model should be discussed in light of the author's conclusions that this may be the best model for CCHF pathogenesis studies. Since the original descriptive paper, the model has also been evaluated by other groups and animals found to have mild disease (Cross et al. PLoS NTD 2020 and Smith et al PLoS Pathogens 2019) along with two papers evaluating an antiviral and vaccine in the model (Hawman et al AVR 2020 and Hawman et al. Nature Micro 2020) that both did not report uniformly severe disease. It is likely that this model most accurately represents human disease which is a full spectrum of disease from asymptomatic to severe disease. However, the variability in the model along with the practical and ethical considerations of NHP the author already presented are significant limitations to using the cyno macaque model for pathogenesis studies or countermeasure development. Author discusses these types of limitations for RVFV (Lines 498 - 502) and similar discussion is warranted here.

Section 4: This section is nearly entirely speculative. Although author lists in table 1 the doses used in many animal studies, for many the LD50 or ID50s of these doses is not reported. Additionally, the titer of a virus is based on tissue culture which may not accurately represent the animal infectious dose. E.g. of 1000 TCID50 injected into an animal, how many particles are captured by non-permissive cells, how many are eliminated by innate responses like natural antibody, and how many viruses actually establish productive infections for a given model and virus? Lastly, it is also unknown what the typical infectious dose is in the human for many of these infections. In the case of EBOV, patients can be shedding massive amounts of virus in bodily fluids leading to large inoculums upon exposure of close contacts. Are there references that describe typical inoculating doses for these viruses? Without this whether these represent actually "high" or "low" challenge doses is speculative.

 Line 649 - 651 "non physiologically relevant doses" or 660-662 "lower, physiologically relevant doses" : Is it accurate to claim this when for many animal models they quite accurately mimic human disease? Or for when the inoculating dose for many of these pathogens is unknown?

Author's statement on line 612 "one of the critical aspects that is often not taken into account is the pathogen dose" or line 675 ", researchers often use the highest dose that is practical, to ensure lethal outcomes without considering the effects that this dose may have in the context o of the experimental goal" is contradicted by author's own statements such as "A high dose is used in many vaccine and therapeutic studies to ensure death in the control group, with incrementally less lethal doses posing the risk of having survivors, which will diminish the statistical power of the performed experiments." (Line 622 - 624). It is an assumption on the author's part that the multitude of referenced studies did not take into account the challenge dose. Rather, it is more likely that these challenge doses were chosen for reasons the author stated.

This section 4 and the author's overall conclusions could be strengthened by discussion of reports where viral dose actually alters viral hemorrhagic fever disease in these mice. For example Haddock et al. (JID 2018) found an inverse correlation between virus dose and survival with mice infected with MA-EBOV. Line 631-632 could be discussed in the context of studies of CCHFV which described non-lethal models and identified protective adaptive immune responses (Hawman et al 2019/2020). Line 640 - 642 could be discussed in the context of the collaborative cross mice and ebola (Rasmussen et al Science 2014 or Price et al Cell Reports 2020). Further I would temper conclusions such as lines 677 - 680 that high doses may reduce the value of these studies. A massive understanding of T-cell biology has been uncovered due to LCMV and mice using mice infected with sometimes as high as 1x10^6 PFU intravenously.

Overall while I agree with the author's sentiments that challenge dose is an important consideration, that higher doses are not always better and that there is utility in non-lethal models, several of the authors statements need to be tempered to reflect the massive understanding of VHF we have gained from these animal models and by the fact that "physiologically relevant" doses for many of these viruses are unknown. Many of these models have been extensively used with extensive characterization of the disease and for many models, they quite accurately represent human disease even with "high" viral doses. The integration of sections 2 through 4 is somewhat weak as section 3 almost has no mention of viral dose then section 4 has almost no references to VHF animal models.

Minor comments:

Line 197: Should 102 and 103 be 10^2 and 10^3?

Line 361 - 363: An additional limitation, arguably more important than those listed is the ethical limitation of using NHPs.

Line 396: Infection should be infect

Line 570: Viruse should be viruses 

Author Response

In the manuscript by Warner et al. the author reviews the effect of antigen dose on development of immune responses, summarizes current animal models for VHF and ends with a discussion on challenge dose and model utility. Overall the paper is well written and has extensive references. However, the later parts of the manuscript are entirely speculative and not well referenced. Although I happen to agree with the author's conclusions that challenge dose is an important consideration for model development and as a further extension that severe but non-lethal models are a useful tool to study immune responses the later portions of the manuscript need revision.

Major comments:

Author focuses on T cells in section 2, what about humoral responses?

I want to thank the author for pointing this out. An additional brief focus on humoral immunity has been added to this section, with a discussion of its importance. However in section 2, there is discussion of antibody responses, specifically in the context of early studies on antigen dose. In addition, the kinetics of antibody responses is depicted in figure 1. While technically, discussions of Th2 responses refers to the phenotype of responding T cells, it is often accepted that that nature of Th2-type responses is defined by humoral immunity, thus discussions in this section of distinct Th1 or Th2 immune responses implies a predominant humoral immune response. This could be made more clear.

With regards to section 3 on CCHFV, the development of an immunocompetent mouse model for CCHFV has been recently reported and should be included in this section.

Thank you to the reviewer for catching this and pointing it out. A reference to this has been included.

Line 453 - 462: The limitations of this model should be discussed in light of the author's conclusions that this may be the best model for CCHF pathogenesis studies. Since the original descriptive paper, the model has also been evaluated by other groups and animals found to have mild disease (Cross et al. PLoS NTD 2020 and Smith et al PLoS Pathogens 2019) along with two papers evaluating an antiviral and vaccine in the model (Hawman et al AVR 2020 and Hawman et al. Nature Micro 2020) that both did not report uniformly severe disease. It is likely that this model most accurately represents human disease which is a full spectrum of disease from asymptomatic to severe disease. However, the variability in the model along with the practical and ethical considerations of NHP the author already presented are significant limitations to using the cyno macaque model for pathogenesis studies or countermeasure development. Author discusses these types of limitations for RVFV (Lines 498 - 502) and similar discussion is warranted here.

I want to thank the reviewer for mentioning these articles and their relevance here. These are valid points and a brief discussion regarding these results has been added to this section.

Section 4: This section is nearly entirely speculative. Although author lists in table 1 the doses used in many animal studies, for many the LD50 or ID50s of these doses is not reported. Additionally, the titer of a virus is based on tissue culture which may not accurately represent the animal infectious dose. E.g. of 1000 TCID50 injected into an animal, how many particles are captured by non-permissive cells, how many are eliminated by innate responses like natural antibody, and how many viruses actually establish productive infections for a given model and virus? Lastly, it is also unknown what the typical infectious dose is in the human for many of these infections. In the case of EBOV, patients can be shedding massive amounts of virus in bodily fluids leading to large inoculums upon exposure of close contacts. Are there references that describe typical inoculating doses for these viruses? Without this whether these represent actually "high" or "low" challenge doses is speculative.

A similar comment was raised by the other reviewer, and I want to thank them for addressing this issue. In addressing the comment from the other reviewer, I have added in some discussion of the issue of “high” vs “low” dose, including how titers are determined etc. as well as issues with trying to compare these with human exposures. I think that including this point is important and I do agree that the section is speculative, though I think that is unavoidable to some degree here due to a lack of data relating to human infections in most cases. I am hoping that, while speculative, the section at least makes readers aware that this might be an issue and should at least be considered, based on how we know dose and route can impact infection outcome, not just for viruses.

 Line 649 - 651 "non physiologically relevant doses" or 660-662 "lower, physiologically relevant doses" : Is it accurate to claim this when for many animal models they quite accurately mimic human disease? Or for when the inoculating dose for many of these pathogens is unknown?

This is a good point brought up by the reviewer. I have made changes to the text in parts referring to physiological relevance in light of this. It may not be the best phrasing to use, when as it is mentioned, that these doses may not be known for many pathogens in human infections.

Author's statement on line 612 "one of the critical aspects that is often not taken into account is the pathogen dose" or line 675 ", researchers often use the highest dose that is practical, to ensure lethal outcomes without considering the effects that this dose may have in the context o of the experimental goal" is contradicted by author's own statements such as "A high dose is used in many vaccine and therapeutic studies to ensure death in the control group, with incrementally less lethal doses posing the risk of having survivors, which will diminish the statistical power of the performed experiments." (Line 622 - 624). It is an assumption on the author's part that the multitude of referenced studies did not take into account the challenge dose. Rather, it is more likely that these challenge doses were chosen for reasons the author stated.

Thank you for catching this contradiction. It did come across that way, in that it was assumed that the authors did not take this into account, when surely it has been taken into account. It likely is often that case that the practicality of using higher doses outweighs other considerations when performing the experiments. I have changed the language in a few of these spots to reflect this.

This section 4 and the author's overall conclusions could be strengthened by discussion of reports where viral dose actually alters viral hemorrhagic fever disease in these mice. For example Haddock et al. (JID 2018) found an inverse correlation between virus dose and survival with mice infected with MA-EBOV. Line 631-632 could be discussed in the context of studies of CCHFV which described non-lethal models and identified protective adaptive immune responses (Hawman et al 2019/2020). Line 640 - 642 could be discussed in the context of the collaborative cross mice and ebola (Rasmussen et al Science 2014 or Price et al Cell Reports 2020). Further I would temper conclusions such as lines 677 - 680 that high doses may reduce the value of these studies. A massive understanding of T-cell biology has been uncovered due to LCMV and mice using mice infected with sometimes as high as 1x10^6 PFU intravenously.

I think that addition of a discussion of these papers is helpful, and have added to the text a brief description of them and how they can be looked at as examples of how dose can be examined in different models. I have altered the language a bit in section five as well. This is in hopes that it does not seem as though I think that all studies using high doses are of lesser value.

Overall while I agree with the author's sentiments that challenge dose is an important consideration, that higher doses are not always better and that there is utility in non-lethal models, several of the authors statements need to be tempered to reflect the massive understanding of VHF we have gained from these animal models and by the fact that "physiologically relevant" doses for many of these viruses are unknown. Many of these models have been extensively used with extensive characterization of the disease and for many models, they quite accurately represent human disease even with "high" viral doses. The integration of sections 2 through 4 is somewhat weak as section 3 almost has no mention of viral dose then section 4 has almost no references to VHF animal models.

I want to thank the reviewer for the important observations and comments regarding the manuscript. Their points have been noted and addressed where possible, and I think that they have improved the manuscript. I do agree that there are many models that have been instrumental in developing our knowledge of disease course for many viruses. The intention was not to diminish or disregard those studies or imply that they have not been valuable. Rather it was to attempt to bring about a discussion regarding a potential limitation in some aspects of animal model usage. I have attempted to temper the language in parts of the text where it may seem like its implied that these studies are not of value. Specifically, the mentions throughout of “physiologically relevant” doses, which I realize will have different meanings in the contexts of different models, viruses, and species, and may not even be known in some cases.  

Minor comments:

Line 197: Should 102 and 103 be 10^2 and 10^3?

This has been updated in the manuscript.

Line 361 - 363: An additional limitation, arguably more important than those listed is the ethical limitation of using NHPs.

Thank you to the reviewer for commenting on this. A statement to this effect has been added to the text.

Line 396: Infection should be infect

Changed in the manuscript.

Line 570: Viruse should be viruses

Updated in the manuscript.

Round 2

Reviewer 2 Report

Author has adequately addressed my comments. Minor spellcheck and grammar check needed.